# Trapping IgE in a closed conformation by mimicking CD23 binding prevents and disrupts FcεRI interaction

Frederic Jabs[1,2], Melanie Plum[1], Nick S. Laursen[3], Rasmus K. Jensen[3], Brian Mølgaard[1], Michaela Miehe[1], Marco Mandolesi [1], Michèle M. Rauber[4,5], Wolfgang Pfützner[4], Thilo Jakob[5], Christian Möbs [4], Gregers R. Andersen[3] & Edzard Spillner [1]

Anti-IgE therapeutics interfere with the ability of IgE to bind to its receptors on effector cells. Here we report the crystal structure of an anti-IgE single-domain antibody in complex with an IgE Fc fragment, revealing how the antibody inhibits interactions between IgE and the two receptors FcεRI and CD23. The epitope overlaps only slightly with the FcεRI-binding site but significantly with the CD23-binding site. Solution scattering studies of the IgE Fc reveal that antibody binding induces a half-bent conformation in between the well-known bent and extended IgE Fc conformations. The antibody acts as functional homolog of CD23 and induces a closed conformation of IgE Fc incompatible with FcεRI binding. Notably the antibody displaces IgE from both CD23 and FcεRI, and abrogates allergen-mediated basophil activation and facilitated allergen binding. The inhibitory mechanism might facilitate strategies for the future development of anti-IgE therapeutics for treatment of allergic diseases.

[1] Immunological Engineering, Department of Engineering, Aarhus University, 8000 Aarhus, Denmark. [2] Institute of Organic Chemistry, Department of Chemistry, University of Hamburg, 20146 Hamburg, Germany. [3] Department of Molecular Biology and Genetics, Aarhus University, 8000 Aarhus, Denmark. [4] Clinical & Experimental Allergology, Department of Dermatology and Allergology, Philipps University Marburg, 35043 Marburg, Germany. [5] Department of Dermatology and Allergology University Medical Center Giessen and Marburg, Justus-Liebig University Giessen, 35385 Giessen, Germany. Frederic Jabs, Melanie Plum, Nick Stub Laursen and Rasmus K. Jensen contributed equally to this work. Christian Möbs, Gregers R. Andersen and Edzard Spillner jointly supervised this work. Correspondence and requests for materials should be addressed to N.S.L. (email: nsl@mbg.au.dk) or to E.S. (email: e.spillner@eng.au.dk)

Allergic diseases can be linked to IgE antibodies present in the circulation and on the surface of a variety of cell types[1]. Although the least abundant type of antibodies, IgE exhibits a variety of structural peculiarities with major functional consequences. IgE acts as a key molecule in a network of proteins, including the high-affinity IgE receptor FcεRI, the low-affinity receptor CD23, and galectins, e.g., galectin-3[2]. Upon crosslinking by allergens, IgE bound to FcεRI on mast cells and basophils triggers degranulation, release of proinflammatory mediators, and immediate reactions[2].

IgE is an evolutionarily conserved and heavily glycosylated heterotetramer (Fig. 1a) with the epsilon heavy chain having four constant domains. The IgE Fc binds to the human FcεRI complex that is expressed as an αβγ2 tetramer or an αγ2 trimer lacking the signal amplifying β-subunit[3–5]. The α-chain of the FcεRI displays an affinity for IgE in the range of $10^{11}$ $M^{-1}$, providing the basis for long-term stability on effector cells and half-life of ~10 days[6].

Strategies to reduce increased levels of IgE and to limit effector cell degranulation included the development of antagonistic anti-IgE antibodies and antibody alternatives including a DARPin and aptamers[7]. The only approved anti-IgE antibody, omalizumab, primarily prevents interaction of free IgE with its receptor on effector cells[8–10] and eventually reverses phenotypic and functional effects of IgE such as enhanced FcεRI levels on effector

cells[11,12]. Not all patients with allergic asthma benefit from treatment[13] and failure may also be caused by pharmacologically active IgE:omalizumab complexes[14] that hamper correct dosing of anti-IgE[15]. Second-generation anti-IgE molecules such as ligelizumab and MEDI4212 are currently under investigation, but initial results suggest limited improvement. Basic structural and functional aspects of anti-IgE, e.g., the mechanism of rapid improvement in chronic urticaria, remain unclear[16,17].

Key for receptor binding and therefore anti-IgE concepts is the IgE Fc that may adopt strongly bent or extended structures with most striking differences in the positioning of the Cε2 domains[18,19]. Furthermore, the Cε3–4 sub-fragment adopts different conformational states ranging from closed to open depending on the spacing of the Cε3 domains and their distance to the Cε4 domains[20]. This conformational flexibility allows the Cε3 domains to rotate (swing) closer to or farther away from each other (Fig. 1b, c).

Structural studies have unraveled how IgE interacts in a highly ordered and specific manner with its receptors. The FcεRI-binding site on the IgE Fc is located primarily on the Cε3 domain whereas CD23 binds to a site involving both the Cε3 and Cε4 domains (Fig. 1a)[21,22]. FcεRI binds to the open and CD23 to the closed conformation of Cε3–4 domains (Fig. 1b) making binding of the two receptors mutually exclusive and preventing overlap of

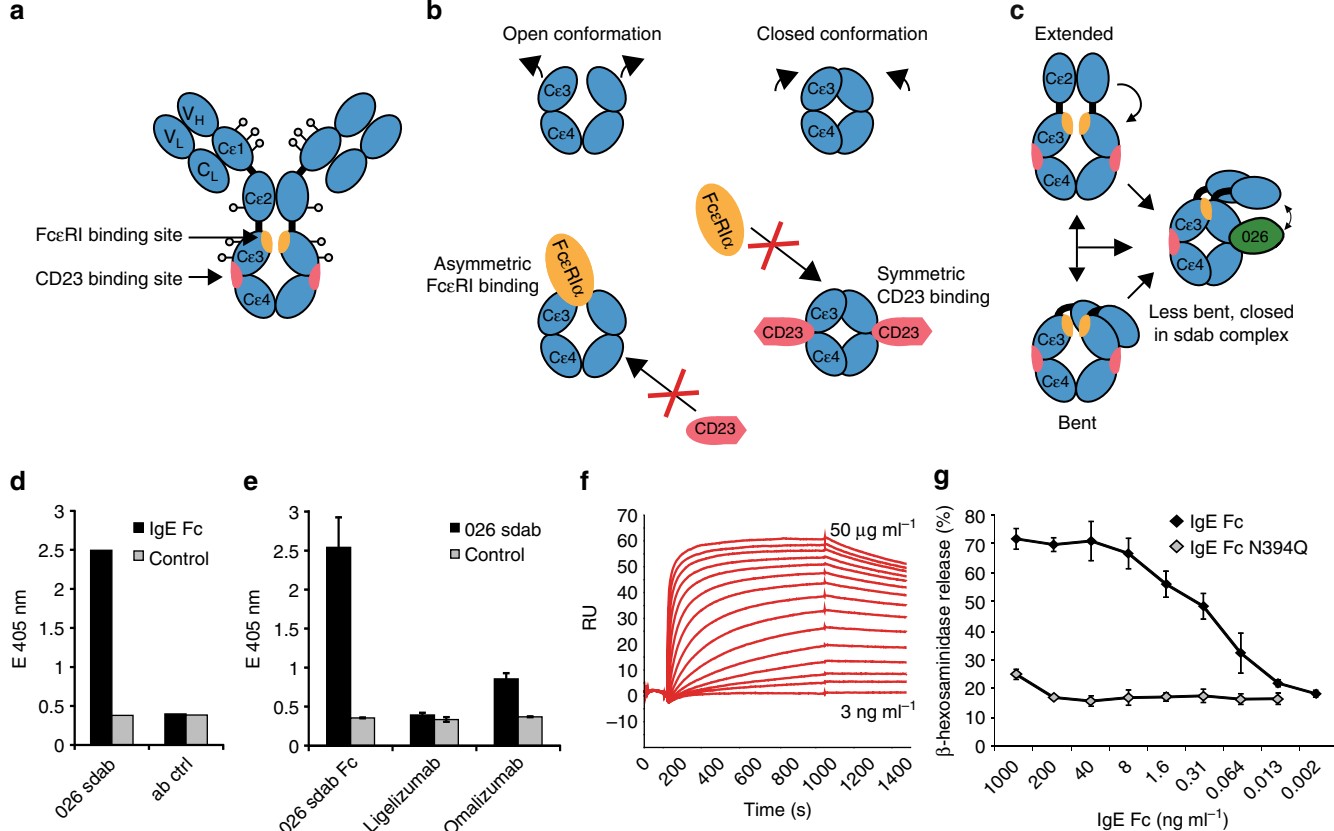

**Fig. 1** Organization and conformational rearrangements of the IgE Fc. **a** IgE and the binding sites of the FcεRI (orange) and CD23 (pink) (adapted from ref. [23]). The glycosylations are indicated by dots. **b** Representation of the open and closed conformations of the IgE Fc Cε3–4 domains, and the mutual allosteric inhibition by FcεRIα (orange) and CD23 (pink). **c** Representation of the bent and extended conformation of IgE Fc Cε2–4 and the conformation in the 026 sdab complex, together with the relative position of the Cε2 domains. **d** Immunoreactivity of the 026 sdab to recombinant IgE Fc was assessed by ELISA. **e** Interference of the 026 sdab with different anti-IgE antibodies was assessed by sandwich ELISA using 026 sdab for capturing IgE Fc. Data are mean ± s.d. Detection of bound anti-IgE antibodies was performed using anti-IgG antibodies coupled to alkaline phosphatase. **f** The affinity of the 026 anti-IgE binding to immobilized IgE Fc was assessed by surface plasmon resonance. **g** Biological activity of recombinant IgE Fc and an IgE Fc lacking the glycan at N394 in mediator-release assays. Data are mean ± s.d. RBL-SX38 cells expressing the human FcεRI were sensitized with IgE Fc. Degranulation was induced by the addition of anti-IgE and monitored by released β-hexosaminidase activity

**Table 1 Data collection and refinement statistics**

| Data collection | |
|---|---|
| Data set | IgE-Fc-026 |
| X-ray source | Diamond I24 |
| Space group | $P4_12_12$ |
| Unit cell parameters | $a = 102.0$, $b = 102.0$, $c = 300.4$ Å |
| | $\alpha = \beta = \gamma = 90°$ |
| Resolution (Å)[a] | 19.96–3.40 (3.52–3.40) |
| Unique reflections[a] | 22,564 (2190) |
| Multiplicity[a] | 12.3 (10.5) |
| $I/\sigma (I)$[a] | 8.3 (1.11) |
| Completeness[a] | 0.99 (1.00) |
| $R_{meas}$[a] | 0.35 (2.37) |
| $R_{pim}$[a] | 0.10 (0.71) |
| $R_{merge}$[a] | 0.34 (2.25) |
| $CC_{1/2}$[a] | 0.99 (0.76) |
| **Refinement statistics** | |
| Reflections in refinement | 22,560 |
| $R_{work}$ | 0.2123 |
| $R_{free}$ | 0.2408 |
| Average $B$-value (Å$^2$) | 111.4 |
| Macromolecule (Å$^2$) | 111.4 |
| Carbohydrates (Å$^2$) | 121.7 |
| Wilson $B$-value (Å$^2$) | 108.2 |
| Protein atoms | 5226 |
| Carbohydrate atoms | 166 |
| **Ramachandran statistics (%)** | |
| Favored | 97.12 |
| Allowed | 2.88 |
| Outliers | 0.00 |
| **RMSD from ideal geometry** | |
| Bond angles (°) | 0.86 |
| Bond length (Å) | 0.004 |
| Clashscore | 3.76 |
| PDB ID | 5NQW |

[a] Values in parentheses are for highest-resolution shell. Statistics were calculated by XSCALE (scaling), phenix.refine (refinement), and MOLPROBITY (validation)

the two pathways. Generally, anti-IgE antibodies are supposed to bind in proximity to the FcεRI-binding site in order to interfere with the receptor binding. A recent study of the omalizumab:IgE complex defined steric interference with FcεRI as the main mechanism for its inhibitory effects[23].

Single-domain antibodies (sdabs) are the antigen-binding moiety of heavy chain antibodies occurring in camelid species and cartilaginous fishes[24,25]. Their small size and peculiar biochemical features render sdabs highly versatile molecules. Here we report the crystal structure and structure in solution of the human IgE Fc in complex with the sdab 026, a llama-derived, humanized sdab that was selected against IgE. Our data show that each IgE Fc is targeted by two sdab having an epitope largely distinct from the FcεRI binding site but overlapping significantly with the CD23-binding site. Inhibition of IgE binding to FcεRI occurs through an allosteric mechanism where the IgE Fc is trapped in a closed, less bent conformation similar to the CD23-bound conformation. Hence, our data provide evidence for a mechanism of inhibition with major implications for the biology and therapeutic targeting of IgE.

## Results
### Structure of IgE Fc bound by two single-domain antibodies.
The 026 sdab, recently developed for IgE targeting in allergic diseases[26], was produced in bacteria and purified from supernatant in yields of up to 80 mg l$^{-1}$. The IgE Fc Cε2–4 was purified

from the supernatant of mammalian cell culture (Supplementary Figure 1a). The proteins exhibited immunoreactivity in ELISA (Fig. 1d). Sandwich ELISA analysis using immobilized 026 sdab as capturing antibody suggested the presence of two 026 sdab binding sites in the IgE Fc as bound IgE could be detected using a bivalent 026 sdab Fc fusion molecule (Fig. 1e). Residual binding of omalizumab to 026 sdab bound IgE Fc suggests a partial overlap of the 026 sdab epitope with the omalizumab epitope. In contrast, the lack of ligelizumab binding might indicate either significant overlap of the epitopes or conformational changes incompatible with ligelizumab binding (Fig. 1e). Surface plasmon resonance analysis using IgE Fc coupled to the sensor surface revealed a dissociation constant $K_D = 1.4$ nM for the sdab, slightly higher than that reported for binding to IgE (Fig. 1f)[26]. Mediator release assays verified biological activity of the IgE Fc and relevance of the glycan at N394 (Fig. 1g).

After purification of the IgE Fc:026 sdab complex (Supplementary Figure 1), we obtained crystals that diffracted X-rays to a maximum resolution of 3.4 Å (Table 1) and determined the structure by molecular replacement. The asymmetric unit contains two copies of the 026 sdab and one IgE Fc molecule. During the process of structure determination it became clear that the crystals contained the IgE Fc Cε3–4 fragment and not the IgE Fc Cε2–4 as expected. Apparently a small fraction of IgE Fc Cε3–4 was sufficient for crystallization of the complex between the sdab and this Fc fragment. Iterative rebuilding and refinement resulted in a final structure with $R_{work}/R_{free}$ values of 0.2123/0.2408 (Table 1). Most residues could be traced with confidence and clear electron density is present for the majority of residues at the sdab:Fc interface (Supplementary Figure 2). The 026 sdab binds symmetrically to the IgE Cε3–4 forming a 2:1 complex where each sdab:Fc interaction buries ~800 Å$^2$ upon complex formation (Fig. 2a, b). The contacts involve surface areas that are clearly different from the FcεRIα binding loops (Fig. 2b, c and Supplementary Figure 3a); thus, the 026 sdab does not inhibit IgE:FcεRI interactions by blocking the binding site. Instead, the sdab is positioned between Cε3 and Cε4 from two different ε-chains, a region also containing the binding site of CD23[22].

As expected the complementarity determining regions (CDRs) in the sdab are responsible for the majority of the IgE contacts. The sdab bridges the Cε3 and Cε4 domains from each ε-chain in the IgE Fc dimer (Fig. 2a, b) with the Cε3 and the Cε4 domains contributing with 30% and 70%, respectively, of the total buried surface area on IgE Fc. Although the resolution of the diffraction data is limited to 3.4 Å, the quality of the electron density map at the sdab:IgE Fc interface allowed us to suggest putative intermolecular hydrogen bonds and salt bridges. The CDR3 is inserted between the Cε3 and Cε4 domains and contacts several residues in the D–E loop of Cε4. Residues D99 and D110 in the sdab possibly engage in electrostatic interactions with the side chain of IgE K497, and E108 appears to make hydrogen bonds to IgE Fc residues 498–501 (Fig. 2c). Several residues in sdab CDR1 and CDR3 may recognize R393 in the Fc C-D loop through hydrogen bonds and salt bridges (Fig. 2d). In addition to the contacts mediated by the CDRs, residues in framework region 2 (FR2) also contribute. A glutamine (Q39) is likely to engage in two hydrogen bonds with the main chain of IgE R440 in the linker between Cε3 and Cε4, and sdab R44 is in position for forming hydrogen bonds to A442 and N468 (Fig. 2e).

In order to validate the crystal structure we mutated residues within sdab CDRs observed to contact IgE (Supplementary Figure 4a–c). Most mutations led to a significant reduction of sdab binding to IgE Fc (Supplementary Figure 4d). In particular, mutation of residues E108, D110 and Y112 in CDR3 abolished IgE binding suggesting a critical role in the interaction and

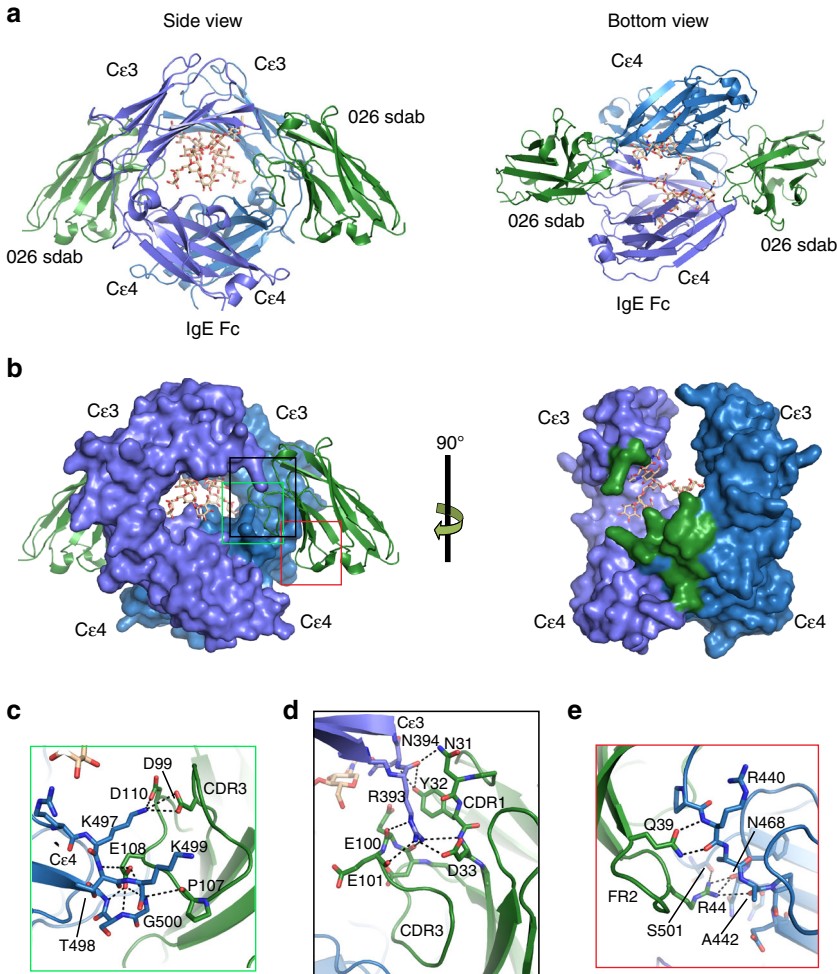

**Fig. 2** The 026:IgE Fc complex and the sdab epitope. **a** A cartoon representation showing IgE Fc Cε3–4 and two 026 sdab molecules binding symmetric sites (green, 026; blue, IgE Fc heavy chain 1; violet, IgE Fc heavy chain 2). The side view of the complex shows the 026 sdab approaching perpendicularly relative to the Cε3–4 domains. A glycan on N394 is located in between the Cε3–4 domains. The bottom view of the complex reveals the two symmetric 026 epitopes on the Cε3 and Cε4 domains. **b** Side view of IgE with bound 026 sdab (left) and without (right). IgE residues in contact with 026 sdab are colored green. **c–e** Detailed view of the IgE Fc:026 sdab interface, with putative hydrogen bonds and salt bridges shown as dotted lines

supporting that the contribution of CDR3 to the interaction is essential.

In IgE, N394 carries an oligomannosidic glycan that is essential for biological activity[27]. In our structure, we observe this as a (*N*-acetylglucosamine)2(mannose)5 (*NAG2MAN5*) hepta-saccharide, which represents the major form in natural IgE derived from myeloma cells as well as allergic patients (Supplementary Figure 5)[28]. The hepta-saccharide has been reported for the structure of free IgE Fc and in both structures the glycans contact the same residues suggesting limited flexibility of the glycan[29]. Despite proximity of the sdab epitope to the glycan there is no direct contact, but water-mediated contacts not visible at this resolution may be present.

**Structural basis for the inhibitory activity of the sdab.** Recently, structures of the Fab fragments of omalizumab[23], two other anti-IgE antibodies[18,30], and a DARPin[31] in complex with IgE Fc domains have been described. All these molecules preferentially bind to the Cε3 domain of IgE critically involved in FcεRIα binding (Supplementary Figure 3). The sdab in contrast exhibits a binding mode clearly different from the DARPin and the Fab fragments, and the IgE residues interacting with sdab CDRs are

also largely distinct from those that engage FcεRIα. Hence, direct competition between the sdab and FcεRI for binding must be negligible (Supplementary Figure 3a). In contrast, there is a significant overlap between the sdab and the CD23 epitopes (Fig. 3a, b). In particular, IgE Fc residues S437-R440 interact with the sdab and also provide an important part of the CD23 epitope (Fig. 3c).

Binding to FcεRI and CD23 stabilizes different conformations of the IgE Cε3–4 domains, the open and closed state, respectively. A direct comparison shows that within the sdab complex IgE Fc adopts a closed conformation similar to that adopted upon CD23 binding, which is incompatible with binding to FcεRI (Fig. 3d and Supplementary Figure 6). The sdab-mediated stabilization of the closed conformation provides a rationale for the sdab's inhibitory mechanism with respect to IgE/FcεRI interaction, whereas CD23 binding to IgE is prevented by steric hindrance.

**Effect of single-domain antibody on FcεRI and CD23 binding.** From the structural data it is evident that the binding sites of FcεRI and the sdab, which was developed as anti-IgE blocking IgE binding to FcεRI, are without significant overlap that could explain an inhibitory activity of the sdab by competition. As expected, the anti-IgE sdab prevented binding of IgE and release

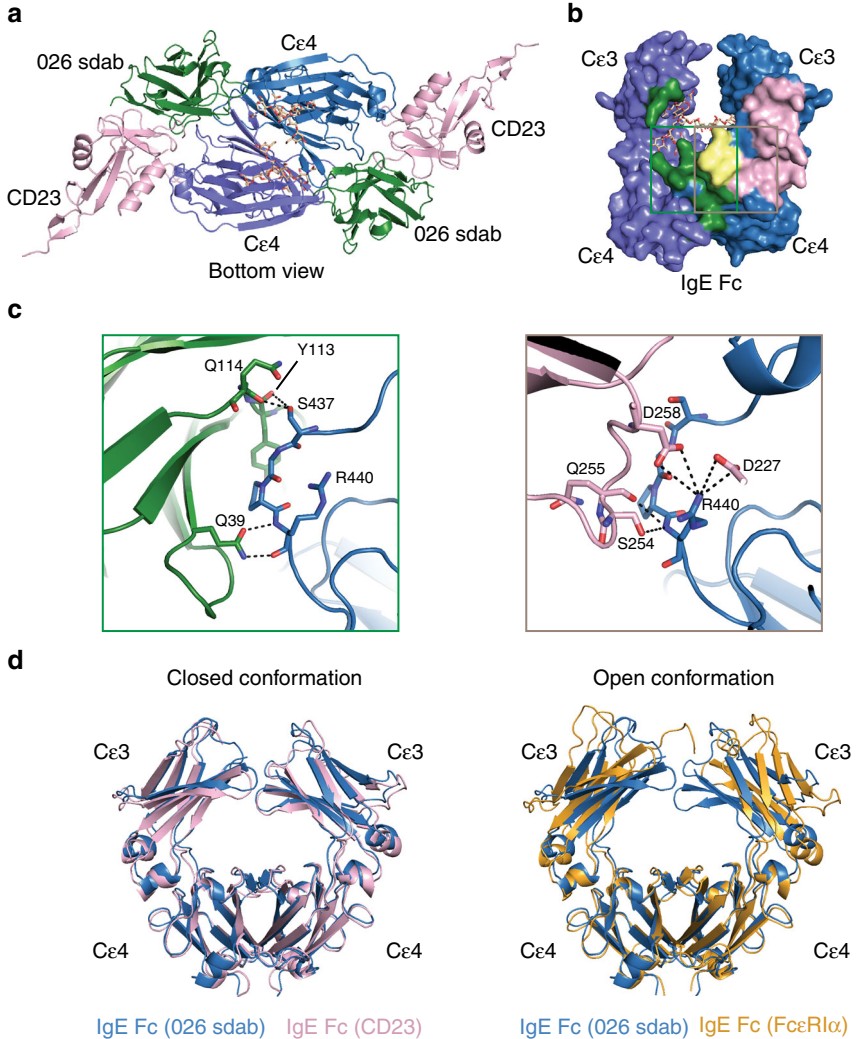

**Fig. 3** Comparison of sdab 026 and CD23 epitopes and IgE Fc conformation. **a** An overlay of the IgE Fc:026 sdab complex with the IgE Fc:CD23 complex shows the proximity of the epitopes of the sdab and CD23. **b** Surface representation of the IgE Fc with the 026 sdab epitope displayed in green, the CD23 epitope in pink and shared epitope residues in yellow. **c** Detailed view of the interface between 026 (left) and CD23 (right, PDB ID: 4GKO) and IgE residues 437–440. Putative hydrogen bonds or salt bridges are shown as dotted black lines. **d** Superposition of the IgE Fc domains reveals the similarity of the IgE Cε3–4 closed conformations in the complex with the sdab in blue and CD23 in pink (left) in contrast to the open conformation in the complex with FcεRI in yellow (right, PDB ID: 1F6A)

of mediators from RBL-SX38 effector cells (Fig. 4a). Recently an anti-IgE DARPin (and omalizumab at higher concentrations) was shown to accelerate dissociation of IgE from FcεRI[23,31]. Therefore we assessed the effect of the sdab on IgE bound to the FcεRI.

Flow cytometry analysis of surface IgE on human basophils obtained from three allergic patients sensitized to inhalative and injected allergen showed that treatment with the 026 sdab for 15 min reduced the amount of surface IgE to ~30% (Fig. 4b, Supplementary Table 1). Prolonging the incubation time further reduced surface IgE to ~20%. Neither the inactive sdab mutant 112 nor omalizumab exhibited comparable reduction. Incubation of immobilized IgE:FcεRI complexes with 026 sdab and omalizumab supported the displacement of IgE by the 026 sdab (Supplementary Figure 7).

Next, we analyzed the impact on basophil activation for a panel of six patients with birch pollen allergy (Supplementary Table 1). The high prevalence of birch pollen allergy and the fact that birch pollen major allergen Bet v 1 is the predominant allergen for the vast majority of birch pollen allergic patients in Northern Europe allows for a good comparability of obtained results. Therefore we

analyzed changes in basophil allergen threshold sensitivity, CDsens, to Bet v 1 in basophil activation tests. CDsens is a well-established measure to quantify effector cell sensitivity to an allergen and reflects the amount of allergen needed for efficient activation. In accordance with the reduced surface IgE, the 026 sdab significantly reduced CDsens for patient's basophils by 50–95%. (Fig. 4c–e). Levels of specific IgE (sIgE) to Bet v 1 and total IgE (tIgE) in serum suggest that lower concentrations of sIgE and as a consequence thereof smaller ratios of sIgE to tIgE translate into a more efficient reduction of CDsens (Supplementary Figure 8). These findings are in line with the observation that polysensitization can be accompanied with a less severe phenotype of the allergic response.

To validate the observed overlap between the sdab and CD23 epitopes, we next assessed the capability of the sdab for interfering with CD23 binding of IgE. ELIFAB assays are able to detect the binding of preformed oligovalent IgE:allergen complexes to CD23[32]. The monovalent sdab has a significantly higher affinity to IgE as compared to monovalent CD23 ($K_d$: 2 μM). Sera of patients with elevated level of specific IgE

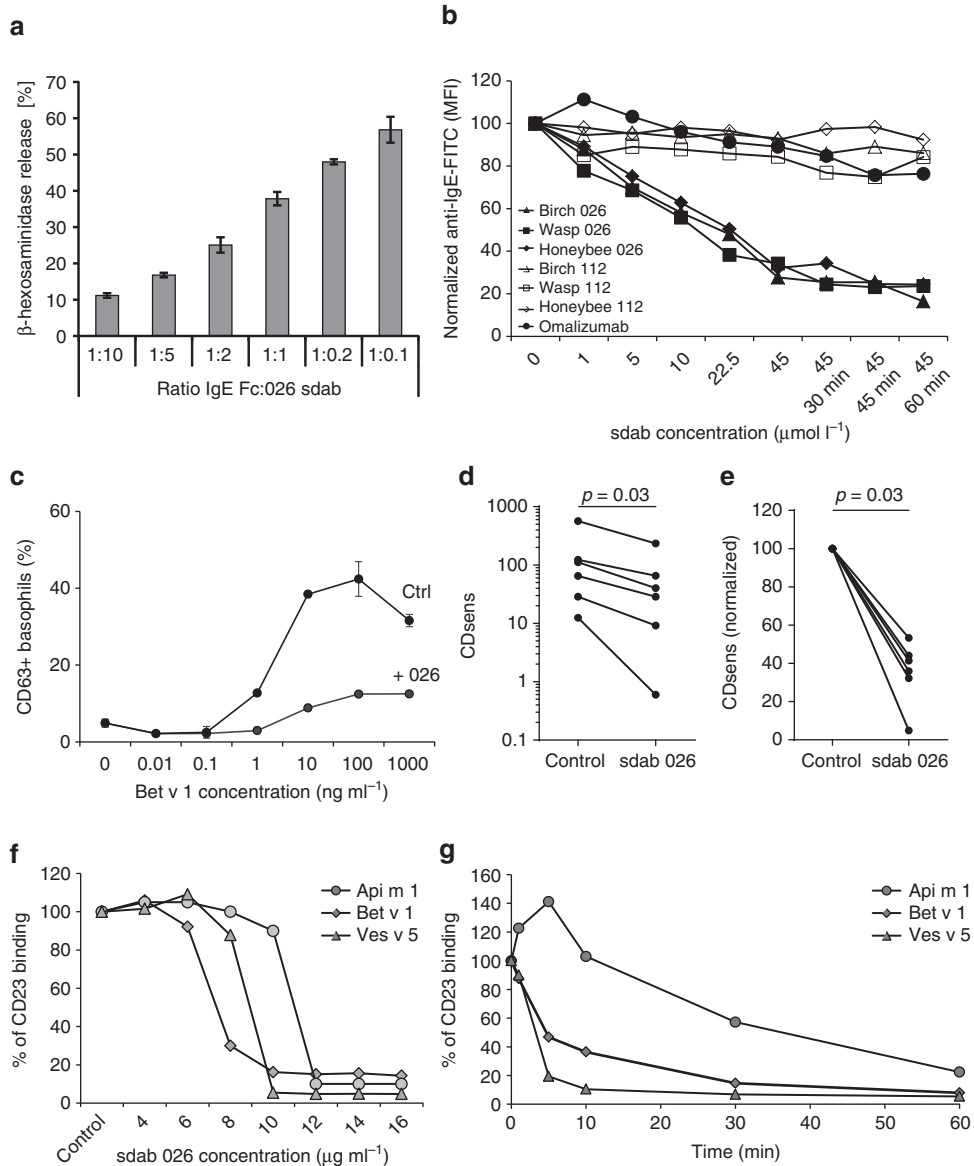

**Fig. 4** Functional activity of the sdab. **a** Inhibition of mediator release from effector cells by 026 sdab was assessed in RBL-SX38 cells providing the human FcεRI. Cells were sensitized with IgE Fc in presence of decreasing amounts of 026 sdab. Degranulation was induced by anti-IgE antibody and monitored by released β-hexosaminidase activity. Data are the mean ± standard deviation of triplicate measurements. **b** Removal of IgE from the surface of basophils by the 026 sdab, the inactive sdab mutant 112 and omalizumab was analyzed by surface staining of IgE using flow cytometry. Basophils of three patients with allergy to different major allergens were used. Concentration of omalizumab was identical to sdab concentrations, but doubled to 90 μmol l$^{-1}$ in the prolonged periods. **c** The capability to reduce basophil sensitivity by displacement of IgE from FcεRI was assessed by basophil activation test (BAT). Basophils were incubated with 026 sdab followed by incubation with Bet v 1. Activation of basophils was then assessed by detecting CD63$^+$ basophils in flow cytometry (exemplary BAT of one donor included in panel **d**). Data are mean ± s.d. **d** The effect of the 026 sdab on basophil activation was analyzed for six birch pollen-sensitized patients. Reduction of effector cell sensitivity was evaluated by CDsens analysis. Comparison of paired samples ±sdab 026 treatment was done by using the nonparametric Wilcoxon signed-rank test. Differences were considered statistically significant at *p* values < 0.05. **e** Inhibition of IgE binding to CD23 by 026 sdab analyzed by ELIFAB. The binding of allergen:IgE complexes to surface bound CD23 was detected by anti-IgE antibodies. Complexes were formed using sera of allergic patients having highly elevated specific IgE to Api m 1, Bet v 1 and Ves v 5 (>100 kUA l$^{-1}$), respectively. **f** The displacement of preformed IgE:allergen complexes from CD23 using sera as in panel **e** by the sdab was analyzed by detecting remaining binding of allergen:IgE complexes after incubation with 026 sdab

(>100 kUA l$^{-1}$) against the honeybee venom allergen Api m 1, the birch pollen allergen Bet v 1 and the major yellow jacket venom allergen Ves v 5 were incubated with the respective allergen and the IgE:allergen complexes applied to CD23. The sdab efficiently prevented binding of the IgE:allergen complexes to CD23 (Fig. 4f). Moreover, even when applied after binding of the complexes to CD23, the sdab displaced IgE efficiently within minutes (Fig. 4g).

**SAXS analysis of single-domain antibody binding to IgE Fc**. We next asked how sdab binding influences the conformation of an IgE Fc including the Cε2 domain. Contaminating IgE Fc Cε3–4 was first removed by hydrophobic interaction chromatography after which the IgE Fc:026 sdab complex was isolated by size-exclusion chromatography (Supplementary Figure 9). Synchrotron small-angle X-ray scattering (SAXS) data were then collected for both unbound IgE Fc and in complex with 026 sdab

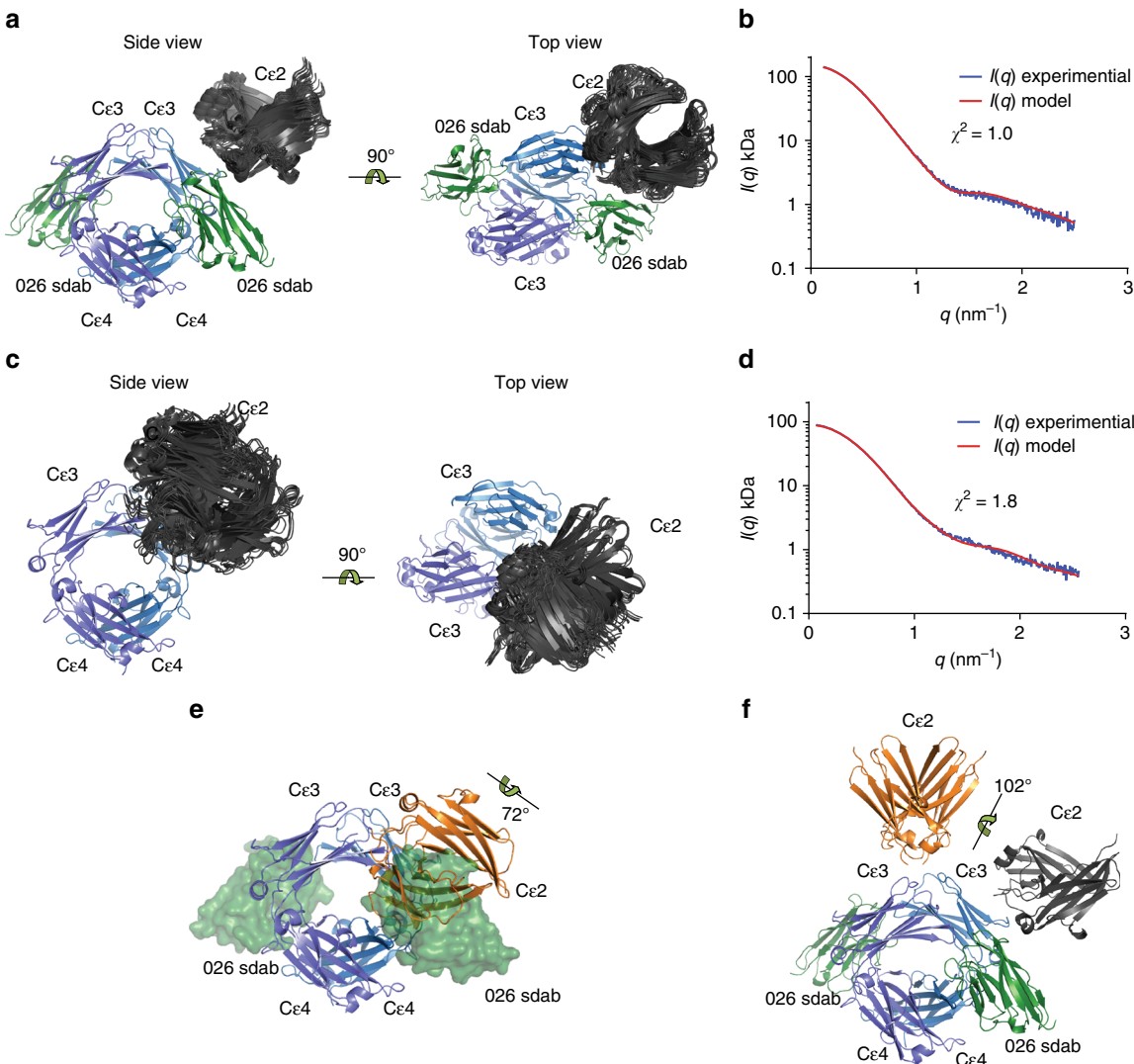

**Fig. 5** SAXS analysis of the IgE Fc:026 sdab complex and unbound IgE Fc. **a** Side and top views of 20 superimposed rigid body models obtained with 026 in green, the Cε3 and Cε4 domains in blue, and the Cε2 domains in gray. **b** The fit of the scattering curve calculated from a representative rigid body model to the experimental curve as output from CORAL. **c, d** Same as panels **a, b**, but for the unbound Fc with the Cε3 domains in the closed conformation. **e** In the fully bent conformation of free IgE Fc (PDB ID: 2WQR) the Cε2 domains (orange) would substantially overlap with the location of sdab 026. **f** Comparison of the extended IgE Fc conformation (Cε2 in orange, PDB ID: 4J4P) and the half-bent conformation of the Fc in complex with sdab 026 observed by SAXS (Cε2 in gray)

(Supplementary Figure 10a–d, Supplementary Table 2). The pair–distance distribution plots showed very similar mass distributions in the two cases but with the IgE Fc:026 sdab complex being slightly larger (Supplementary Figure 10e). Rigid body modeling of the IgE Fc:026 sdab complex with the Cε3 domains fixed in the closed conformation as in our crystal structure, resulted in a tightly clustered ensemble of solutions fitting the SAXS curve well with an average $\chi^2 = 1.06 \pm 0.04$ according to CORAL (Fig. 5a, b). Likewise, we conducted rigid body refinement against the SAXS data collected from unbound IgE Fc with the Cε3 domains fixed in either the closed or the open conformation during refinement. Both scenarios resulted in solutions fitting the data with $\chi^2$ values in the range 1.8–2.3 according to CORAL. Qualitatively output models from both scenarios were in the bent conformation with the Cε2 domains located toward the Cε3 domains (Fig. 5c, d). In roughly 95% of the output models with the Cε3 domains fixed in the closed conformation, a substantial overlap between the Cε2 domains and a bound sdab 026 would occur. In agreement with this, a comparison of our crystal structure of the IgE Fc:026 sdab complex not containing the Cε2 domains with the crystal structure of unbound IgE Fc (pdb ID: 2WQR)[21] reveals that in the sdab complex, IgE Fc must adopt a significantly less bent conformation than that observed in the crystal structure of unbound IgE Fc to avoid overlap between the Cε2 domains and bound sdab (Fig. 5e). Hence, the rigid body modeling provided evidence for a sdab-mediated displacement of the Cε2 and unbending of the IgE Fc resulting in a so far undescribed conformation of the Fc Cε2–4 domains. But an ~100° rotation would still be required to reach the fully extended conformation of IgE Fc (Fig. 5f).

To investigate whether the SAXS data for IgE Fc and IgE Fc:026 sdab complex could be better explained by a mixture of conformations, we also employed the ensemble optimization method (EOM)[33]. For the IgE Fc:026 sdab complex the selected structures clustered around a higher $D_{max}$ than the average generated structure (Supplementary Figure 11a). The two representative structures adopted different bent conformations. The conformation representing 63% of the scattering was almost

identical to the rigid body structures output from CORAL, whereas the second conformation was slightly more extended. The ensemble fitted the experimental data with $\chi^2 = 1.27$ as calculated by GAJOE (Supplementary Figure 11b). The corresponding EOM analysis of the unbound IgE Fc SAXS data resulted in ensemble optimized structures falling in two separate clusters with either a significantly smaller $D_{max}$ or a significantly larger $D_{max}$ than the average generated structure (Supplementary Figure 11c). The ensemble fitted the experimental data with $\chi^2 = 2.43$ (Supplementary Figure 11d). The low $D_{max}$ cluster accounting for ~70% of the scattering corresponded to a bent conformation similar to the CORAL rigid body solutions, whereas the high $D_{max}$ cluster corresponded to a conformation falling in between the fully extended and bent and accounted for ~30% of the total scattering. Overall, the EOM analysis of the solution scattering data confirmed the models obtained by rigid body refinement but in addition provided evidence of a second conformation in both cases. Altogether our SAXS analysis suggested that IgE Fc in solution mainly adopts the bent conformation as anticipated and that binding of 026 sdab induces IgE Fc unbending to a conformation that is in between the classical bent and the fully extended two-fold symmetric conformation.

## Discussion

Key to intervention in allergic diseases is the reduction of IgE in the serum and eventually on effector cells. In this study, we show that the anti-IgE sdab 026 targets IgE by binding to an epitope within the Fc domains that does not significantly overlap with the FcεRI-binding site. Notably the epitope is a conformational epitope involving two domains and a buried surface of ~800 Å[2] that is larger than the omalizumab epitope[23]. Both epitope and inhibition mode of sdab 026 are different from those previously observed for anti-IgE molecules. The sdab blocks interaction with FcεRI by trapping IgE Fc in a closed conformation of the Cε3 domains incompatible with FcεRI binding and a slightly more extended conformation of the Cε2 domains.

The sdab epitope shows significant overlap with the CD23-binding site. Although additional importance of the stalk region of CD23 in binding to IgE has been postulated[34] the shared motif of the sdab and CD23 epitopes and the high affinity of the 026 sdab render an effect of the stalk on the inhibitory activity of the sdab unlikely. As observed in the ELIFAB analyses the inhibition of IgE binding to CD23 and displacement of IgE from CD23 strongly supports the epitope inferred from the structure and the mode of action of the anti-IgE sdab. Recently it has been verified that omalizumab also blocks the interaction with CD23[34], most likely due to minor overlap of the epitope with the binding site[23].

Binding of allergen:IgE complexes to CD23, a C-type lectin and low affinity IgE receptor, is crucial for facilitated antigen presentation and transport across the epithelium, and amplifies the generation and epitope spread of specific IgE[2,35]. The inhibition of CD23 binding by patients' serum IgG correlates with the activity of blocking IgG induced by successful allergen immunotherapy[36] and leads to reduced T-cell activation[37]. Combining immuno-therapy with omalizumab results in clear benefits for treatment efficacy[38]. The main target of anti-IgE approaches however remains interference with FcεRI-mediated immediate effects.

IgE Fc adopts conformational states within the Cε3–Cε4 domain pairs ranging from an open conformation when in complex with FcεRI to the closed conformation when bound to CD23. Binding of FcεRI and CD23 to IgE is mutually exclusive as each is dependent on the conformation of Fc. FcεRI binding to

free IgE moves the two Cε3 domains away from each other, decreases the angle between Cε3 and Cε4 and makes the binding site for CD23 inaccessible. Conversely, binding of CD23 to free IgE increases the angle between Cε3 and Cε4, moves the Cε3 domains to a more closed conformation and prevents IgE Fc interactions with FcεRI. In complex with the sdab, the Cε3 and Cε4 domains adopt the closed conformation, which is incompatible with FcεRI binding. CD23 is contacting residues in Cε3 and Cε4 within the same chain and enforces the closed conformation by pushing the Cε3 domains closer together. With a similar result, the sdab binds Cε3 and Cε4 from different ε chains forcing a closed conformation by pulling the Cε3 from one chain toward the other.

Most of the structures available for IgE Fc contain only the Cε3–4 domains, and likewise we did not obtain crystals with the entire IgE Fc probably since the flexibility of the Cε2 domains in the IgE Fc hampers crystallization. The Cε3–4 fragment is often used in a stabilized form having artificial disulfide bridges or reduced glycosylation as a result of mutagenesis or choice of the expression host. Here we use the authentic IgE Fc and thereby minimize the risk of trapping the molecules in a rare or artificial conformation.

Being unable to crystallize sdab in complex with the full IgE Fc we used SAXS to analyze the complex and free IgE Fc in solution. This allowed insights into the positioning of the Cε2 domain in solution for both the free Fc as well as in the complex with the sdab. Two different approaches to SAXS based modeling suggested that with respect to the sdab complex, the IgE Fc Cε2 domains appear to adopt a primary conformation in between the two well-established bent and extended IgE Fc conformations. In unbound IgE Fc, the dominating conformation appears to be the bent conformation. In addition, EOM analysis suggested that one minor conformation somewhat different from the major conformation may be present in both cases.

Recently, the displacement of IgE from FcεRI by a DARPin IgE antagonist was described[23,31], and a similar, but lower activity has been reported for omalizumab. Our data suggest that the 026 sdab displaces IgE from FcεRI more efficiently than omalizumab. An even four-fold molar excess of omalizumab Fab fragment over the sdab did not result in significant IgE displacement. Apparently, the sdab removes IgE from the surface of effector cells as shown for basophils obtained from patients with different types of allergies. The reduction of IgE correlates with a decreased sensitivity of the effector cells to allergen-dependent activation as shown in a cohort of birch pollen-sensitized patients. It is imperative to consider that sensitivity of effector cells is shaped by the characteristics of surface IgE, e.g., ratio of specific and total IgE, affinity and repertoire complexity[39]. Although we used the dominant birch pollen allergen, Bet v 1, patient-specific sensitization profiles influence the reduction of cellular sensitivity, an effect likely to be even more pronounced in multisensitized individuals.

With respect to the accessibility of one sdab-binding site on the IgE even when bound to the FcεRI, the disruptive activity of the sdab is most likely driven by the conformational rearrangement upon sdab binding to the FcεRI bound IgE. With the capability of the E2_79 DARPin, omalizumab and the 026 sdab for displacing IgE it appears reasonable that the majority of anti-IgE molecules might exhibit similar activity, at least to a certain extent. Increasing this activity could become an important strategy for improving efficacy of anti-IgE therapeutics[40]. Although the removal of IgE from both circulation and effector cells could represent a benefit, the in vivo effect has not been fully proven yet. It has been shown that reduction of IgE on effector cells is counterbalanced by increased sensitivity[41]. Additional studies are therefore needed to address the displacement of IgE and its consequences and implications in detail.

Single-domain antibodies exhibit high production yield in simple expression systems and extraordinary stability, properties that render them interesting for numerous biotechnological and biomedical applications[42]. Targeting IgE with sdabs could offer a variety of benefits. Their 10-fold smaller size as compared to IgG allows efficient targeting of less accessible sites and the monovalent format prevents formation of larger complexes. State-of-the-art strategies for half-life extension and multiple targeting can easily be applied to sdabs. Delivery in functional form via mucosal and airway tissues is possible using sdabs[43] and could improve the use of difficult routes for anti-IgE application[44].

The sdab's mode of action identified in this study could also open up for the development of novel anti-IgE molecules of even lower molecular mass. It seems reasonable that high affinity targeting of an epitope similar to the sdab epitope by smaller molecules might drive similar conformational rearrangements. Thus, our description of the 026 sdab mode of action is likely to accelerate the development of anti-allergy and asthma drugs in the future.

## Methods

**Crystallization.** The 026 sdab:IgE Fc complex was concentrated to 5 mg ml$^{-1}$ in 20 mM HEPES, 50 mM NaCl, pH 7.2, and crystallized by vapor diffusion in sitting drops formed by mixing protein and reservoir solution containing 0.1 M imidazole pH 7.0, 11–12% polyethylene glycol (PEG) 20,000 in the ratio 1:1. Crystals were cryocooled in liquid nitrogen after transfer to a cryo-protectant composed of 0.1 M imidazole pH 7.0, 12% w/v PEG 20,000 and 30% glycerol.

**Data collection and structure determination and refinement.** Data were collected at Diamond I24[45] and processed with XDS[46]. The structure was determined by molecular replacement with PHASER[47] using the pdb ID: 2WQR as search model[21]. The model was rebuild in Coot[48] and refined with Phenix.refine and iMDFF[49,50]. Figures were prepared with the PyMOL Molecular Graphics System (Schrödinger LLC).

**Protein expression and purification.** The IgE Fc constant region cDNA was obtained from an IgE expression vector initially constructed from a human cDNA library. The cDNA was introduced into an expression vector providing a human immunoglobulin signal sequence via SmiI and MssI restriction enzymes[51]. Point mutations were introduced by PCR Human embryonic kidney cells (HEK-293, ATCC) were cultivated in Dulbecco's modified Eagle medium (DMEM) supplemented with 10% (v/v) heat-inactivated fetal calf serum (FCS), 100 IU ml$^{-1}$ penicillin and 100 µg ml$^{-1}$ streptomycin. HEK-293 cells were transfected with the expression vector using Nanofectin (GE Healthcare) according to the recommendations of the manufacturer. After selection by addition of zeocin stably transfected cells showed an expression yield of 10–15 mg l$^{-1}$. Cell culture supernatant obtained in roller flasks (Greiner) was collected and subsequently subjected to purification of the IgE Fc via nickel based affinity chromatography. The supernatant was loaded on a 1 ml HisTrap excel column (GE Healthcare) equilibrated with PBS (500 mM NaCl, 40 mM Na$_2$HPO$_4$, 10 mM NaH$_2$PO$_4$, pH 7.4). After washing with 10 column volumes (CV) of PBS and 20 CV of 5% PBS/imidazole (100 mM NaCl, 40 mM Na$_2$HPO$_4$, 10 mM NaH$_2$PO4, 300 mM imidazole, pH 7.4) the IgE Fc was eluted in a 5–100% gradient of PBS/imidazole. Pooled fractions were dialyzed against Buffer A (20 mM NaOAc, 50 mM NaCl, pH 6.5) and subsequently applied to a 1 ml Mono S 5/50 column (GE Healthcare) equilibrated with Buffer A. After washing with Buffer A, the protein was eluted with 20 CV of Buffer B (20 mM NaOAc, 1 M NaCl, pH 6.5) with a 0–50% gradient. The IgE Fc was further purified by size exclusion chromatography using a 24 ml Superdex 200 20/300 GL column (GE Healthcare) equilibrated in 20 mM HEPES, 50 mM NaCl, pH 7.2.

The DNA of the sdab 026 (patent WO 2012′/175740 A1) was obtained as a synthetic gene and cloned into the bacterial expression vector pET22+ providing a pelB signal sequence and a C-terminal histidine tag (Supplementary Table 3). Expression of the sdab 026 and its variants was performed for 3 h at 30 °C after IPTG induction. The supernatant was subjected to nickel-based affinity chromatography on a 1 ml HisTrap excel column (GE Healthcare) equilibrated with PBS. After washing with 10 CV of PBS and 20 CV of 5% PBS/300 mM imidazole the protein was eluted with a 5 CV gradient from 5–100% of PBS/imidazole. The sdab mutants were purified accordingly. After dialysis against Buffer A (20 mM Tris, 50 mM NaCl, pH 8.5) the sdab was further purified using ion exchange chromatography using a 6 ml Resource Q column (GE Healthcare) equilibrated in Buffer A. After washing with 5 CV of Buffer A, the protein was eluted with a 5 CV gradient from 0–50% Buffer B (20 mM Tris, 1 M NaCl, pH 8.5).

Prior to SAXS data collection IgE Fc was further purified to remove the Cε3–4 fragment by hydrophobic interaction chromatography using a 1 ml Source 15PHE column (GE Healthcare) equilibrated in Buffer A (1.6 M NH$_3$SO$_4$, 20 mM HEPES,

pH 7.2). The sample obtained from size exclusion chromatography was dialyzed against Buffer A and applied to the column. After washing with 5 CV of Buffer A, the protein was eluted with a 15 CV gradient from 0–100% Buffer B (50 mM NaCl, 20 mM HEPES, pH 7.2). IgE Fc Cε2–4 from the Source 15Phe purification was mixed with purified 026 sdab and the complex was purified by size-exclusion chromatography as described above.

**Characterization of IgE Fc and 026.** The binding of 026 sdab to IgE Fc was assessed in ELISA. Purified IgE Fc (50 µg ml$^{-1}$) was coated on microtiter plates (Greiner) at 4 °C and blocked with 40 mg ml$^{-1}$ milk powder in PBS. Thereafter, 026 sdab at a concentration of 10 µg ml$^{-1}$ was incubated for 2 h at RT. After washing, binding to IgE Fc was detected using alkaline phosphatase-conjugated anti-human kappa antibody and anti-human IgG antibody diluted 1:30,000 and 30 µl of substrate solution.

For comparative assessment of anti-IgE antibodies, ELISA microtiter plates were coated with 026 sdab (25 µg ml$^{-1}$) at 4 °C and blocked with 40 mg ml$^{-1}$ milk powder in PBS. After incubation with IgE Fc, the anti-IgE antibodies were incubated in a final volume of 20 µl for 4 h at room temperature. After washing bound antibodies were detected using an alkaline phosphatase-conjugated anti-human IgG antibody diluted 1:20,000, and 30 µl of substrate solution (5 mg ml$^{-1}$ 4-nitro-phenylphosphate, AppliChem).

**Affinity measurements.** The affinity of the sdab was determined by using a T200 Biacore system. IgE Fc was immobilized to a total of 110 resonance units (RU) onto a CM5 sensor chip (GE Healthcare) using NHS/EDC coupling procedures. Analyses were performed at 25 °C in a buffer containing 10 mM monosodium phosphate, 40 mM disodium phosphate and 100 mM NaCl, pH 7.4 supplemented with 0.01% Tween-20. For kinetic analyses the 026 sdab was injected at concentrations from 0.003 to 50 µg ml$^{-1}$ at a flow rate of 25 µl min$^{-1}$. Association and dissociation were assessed for 600 s. Regeneration of sensor surfaces was performed by subsequent injection of 1 M Tris buffer, pH 10. The dissociation constant at equilibrium $K_D$ was calculated using a 1:1 binding model and the Biacore T200 evaluation software.

**Basophil activation tests and mediator release assays.** Peripheral EDTA blood from allergic donors was preincubated with 026 sdab, mu112 or omalizumab (Novartis) at concentrations from 0 to a maximum of 45 µmol l$^{-1}$ (sdab 026 and mu112) and from 0 to 90 µmol l$^{-1}$ (omalizumab), respectively, in IL-3-containing stimulation buffer (Bühlmann) for up to 60 min on a shaker. Samples were stained with anti-CCR3-PE in a 37 °C water bath for 15 min and anti-IgE-FITC (both BioLegend) for 20 min at 4 °C. Erythrocytes were lysed for 7 min followed by centrifugation at 500 × g for 5 min. Cell pellets were resuspended in 100 µl washing buffer and the mean fluorescence intensity (MFI) of IgE on CCR3$^+$SSC$^{low}$ basophils was measured by flow cytometry (FACS Calibur, BD Biosciences).

To analyze the impact of IgE026 on basophil activation, peripheral EDTA blood from six patients with birch pollen allergy was incubated ±45 µmol l$^{-1}$ sdab 026 for 60 min on a shaker. Afterwards, Bet v 1 (Biomay) diluted at different concentrations (0.01–1000 ng ml$^{-1}$) in stimulation buffer was added. In each case one sample was left unstimulated to exclude background activation. Next, all samples were incubated with 15 µl of staining reagent (Bühlmann) consisting of anti-CCR3-PE and anti-CD63-FITC in a 37 °C water bath for 15 min. As above, erythrocytes were lysed for 7 min followed by centrifugation at 500 × g for 5 min. Cell pellets were resuspended in 100 µl washing buffer and basophil activation was measured by flow cytometry (FACS Calibur, BD Biosciences). Basophils were identified as CCR3$^+$SSC$^{low}$ cells and CD63$^+$ basophils were considered as activated. All results were analyzed using BD FACSDiva software (BD Biosciences).

CDsens analyses were performed as basophil allergen threshold stimulation. Basophil allergen sensitivity was measured as the allergen concentration eliciting 50% (EC50) of maximum CD63 upregulation. The CDsens is defined as inverted value for EC50 multiplied by 100, and was calculated by the following formula: CDsens = 1/EC50 × 100[52].

In order to analyze in vitro degranulation RBL-SX38 cells (kindly provided by J.P. Kinet) were sensitized with rIgE Fc[53]. After washing with incomplete Tyrode's buffer, receptor cross-linking was performed by incubation with goat anti-human IgE for 60 min at 37 °C. Release of β-hexosaminidase from viable vs. lysed cells was assessed for 60 min at 37 °C using p-nitrophenyl N-acetyl-glucosaminidine (Sigma-Aldrich) as a substrate. After stopping the reaction by addition of carbonate buffer (0.1 M, pH 10) the absorbance was recorded at 405 nm.

**Analyses of CD23 binding using the ELIFAB assay.** In order to evaluate the capability of 026 sdab to inhibit the binding of IgE:allergen complexes to CD23 the ELISA-based ELIFAB assay, a cell-free variant of the FAB assay, was performed[32,54]. To allow for formation of IgE:allergen complexes 20 µl of indicator serum containing either high Api m 1-, Bet v 1- or Ves v 5-specific IgE concentrations (all >100 kUA l$^{-1}$) was preincubated with the respective allergen at 37 °C for 1 h in the presence of different 026 sdab concentrations. Following preincubation, IgE:allergen complexes were transferred to plates coated with CD23 (R&D Systems, Bio-Techne) and incubated for 1 h at RT. IgE:allergen complexes bound to immobilized CD23 were detected by adding biotin-conjugated anti-

human IgE antibody (BD Biosciences), streptavidin-peroxidase (Sigma-Aldrich), and 3,3′,5,5′-tetramethylbenzidine (TMB) substrate (Calbiochem, Merck Millipore). All samples were analyzed in duplicates.

In a second set of ELIFAB experiments, preformed IgE:allergen complexes bound to CD23 were incubated with 5 μl 026 sdab up to 60 min evaluating the ability of 026 sdab to displace IgE:allergen complexes from CD23. Displacing IgE:allergen complexes from immobilized CD23 was determined at a 026 sdab concentration of 12 μg ml$^{-1}$ for Api m 1 (Latoxan), 8 μg ml$^{-1}$ for Bet v 1 (Biomay) and 11 μg ml$^{-1}$ for Ves v 5, respectively.

**Small-angle X-ray scattering data collection and modeling**. The SAXS measurements of IgE Fc and IgE Fc in complex with sdab 026 were performed in batch mode at the BM29 beamline at the European Synchrotron Radiation Facility (ESRF), Grenoble, France[55]. The data were collected using a PILATUS 1 M pixel detector and $\lambda = 0.992$ Å in a temperature controlled capillary at 4 °C. The sample-to-detector distance was 2.872 m, covering a range of momentum transfer $0.04 < q < 5$ nm$^{-1}$ ($q = (4 \times \pi \times \sin\theta) \div \lambda$, where $2\theta$ is the scattering angle). Samples in 20 mM HEPES, 50 mM NaCl, pH 7.2 were investigated in the concentration ranges 0.5–1.7 and 2.2–9.0 mg ml$^{-1}$ for the IgE Fc:026 sdab complex and IgE Fc, respectively. Data were collected with 10 exposures of 2 s. Radial averaging, buffer subtraction and concentration scaling were performed using the beamline pipeline[56]. Bovine serum albumin (Sigma, A7638-5GP) used for calculation of the molecular weight (Supplementary Table 1) was solubilized in a buffer containing 50 mM Hepes pH 7.5 at concentration of 5 mg ml$^{-1}$. For the IgE Fc:026 sdab complex a concentration of 1.7 mg ml$^{-1}$ was used for further modeling. For IgE Fc the measurements at 2.2 and 9.0 mg ml$^{-1}$ were merged using ALMERGE[57]. The pair distribution function was calculated by indirect Fourier Transform using GNOM[58]. Rigid body refinements were performed using a momentum transfer range of $q < 3.0$ nm$^{-1}$ with CORAL[59]. The 026 sdab and the Fc Cε3 and Cε4 domains were used as a single rigid body taken directly from our crystal structure. The two Cε2 domains from the pdb entry 2WQR were grouped into a single rigid body, and the C-terminal of both of the Cε2 domains were linked to the N-terminal residue of their corresponding Cε3 domain with a distance restraint of 25 Å. Calculation of the angle of rotation between the SAXS models and the extended (RCSB: 4J4P) and bent (RCSB: 2WQR) conformation of IgE Fc was performed using DynDom[60]. Ensemble optimization of both unbound IgE Fc and the IgE Fc:026 sdab complex was performed using the EOM 2.0 programs RANCH and GAJOE[33]. A total of 5000 models were generated utilizing the same rigid bodies as for CORAL rigid body modeling, but with a randomly generated alpha carbon trace connecting the two rigid bodies, instead of a distance restrain. GAJOE was then used to select the ensemble of models best representing the scattering curve.

**Patient material**. The study was approved by the Ethics Committee of the Medical Faculty of Philipps University, Marburg, Germany; all patients provided written informed consent to participate in the trial.

**Data availability**. Coordinates and structure factors have been deposited in the Protein Data Bank under accession code 5NQW. Other data are available from the corresponding authors upon reasonable request.

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

## Acknowledgements

We acknowledge the help of Jesper Lykkegaard Karlsen with refinement and Manuel Schulze-Dasbeck for expert technical assistance. We are grateful to local contact at the ESRF for providing assistance in using beamline BM29. The authors would like to thank Diamond Light Source for beamtime, and the staff of beamlines I02, I03, and I24 for assistance with crystal testing and data collection. This work was supported in part by a research grant of the Danish Council for Independent Research to E.S. G.R.A. was supported by the Danish Council for Independent Research, the Lundbeck foundation, and Danscatt, N.S.L. was supported by the Lundbeck Foundation.

## Author contributions

The study was conceived by F.J., M.P., N.S.L., R.K.J., C.M., G.R.A., and E.S. F.J., M.P., N.S.L., R.K.J., B.M., M.M., and Ma.Ma. participated in all stages of the project and performed the experiments. G.R.A., N.S.L., and E.S. wrote the manuscript together with F.J., M.P., N.S.L., C.M., T.J., and R.K.J. C.M., M.M.R., and W.P. provided and analyzed the patient material. All authors contributed to the interpretation of the data and provided critical review of the manuscript.

## Additional information

**Competing interests:** The authors declare no competing financial interests.

