## [Peer Review File · Nature Communications]

Reviewers' comments:

Reviewer #1 (Remarks to the Author):

This is an exciting study where the authors have characterised a single domain antibody which displaces IgE from both high and low affinity IgE receptors. The mode of action is different from existing technologies which reduce IgE-IgE receptor binding (Omalizumab & DARPin's) and appears to be considerably more effective at displacing IgE in comparison. The manuscript is very well written and I have no major concerns with most of the work shown. However, there does not seem to be any data regarding the kinetics of disassociation of IgE from FcεRI due to the effects of the single domain antibody. Another issue is that the data shown regarding human basophil tests in figure 4 seem to be derived from only one patient and none of these functional assays show relevant controls or statistics. Given differences in IgE sensitization and the functional sensitivity of basophils from donor-to-donor it would be useful to quantify the efficacy and potency of the single domain antibody at inhibiting IgE-dependent functional responses and IgE sensitization over a range of basophil donors.

Minor point:

Page 11, line 8: "CD23 contacts residues in...." Change to "contact residues"?

Reviewer #2 (Remarks to the Author):

This manuscript describes the crystal structure of an IgE Fc fragment in complex with a single domain antibody, as well as the structure in solution as determined by solution scattering. These structural results as well as biochemical studies help to explain the mechanism of inhibition of the single domain antibody. The manuscript is clear and well written and the results should be of interest for anyone interested in antibody-antigen interactions or in therapeutic antibodies.

Specific comments:

p. 4, in the introduction please describe sdab 026- was it from a camel or llama, or is it a human VHH? How was it isolated?

p. 6, please give a better description of the 'hepta-saccharide'- was this a Nag2Man4?

p. 13, in the materials and methods, hopefully your crystals were 'cryocooled', not frozen.

In Table 1, please include Rmerge and Rpim. Under the refinement statistics, what are the ligand atoms? I would specify as water or carbohydrate or whatever they are to be more clear.

In the Figure 3 legend, please include the PDB code for the Fc-FcεRI complex shown in panel D.

Reviewer #3 (Remarks to the Author):

In "Trapping IgE in a closed conformation by mimicking CD23 binding prevents and disrupts interaction with FcεRI" Jabs et al. describe a novel inhibitory mechanism for binding of IgE to relevant receptors. This review focuses on the SAXS analysis presented in the manuscript.

Foremost, I would like to thank the authors for an interesting read. There are some minor issues that can be resolved by either adjustments of the text or straightforward additional modeling.

Parts of the introduction read rather awkwardly, it might profit from thorough proofreading by an

outsider. For example in "The human FcεRI is expressed as a tetramer or a trimer lacking the signal amplifying β-subunit." It is not clear whether only the trimer or both the trimer and the tetramer lack the subunit.

Page 5, 1st paragraph, analysis of the surface plasmon resonance experiment: The combination of a KD value with a χ^2 is nonsensical. The KD should have a margin of confidence, whereas χ^2 is a property of the fit (and often meaningless).

Page 9, SAXS analysis: It is not possible to draw any conclusion of the ensemble of solution structure from CORAL modeling. Clustering of the results only imply that the reconstruction is stable, but CORAL by design will always find the SINGLE model that describes the data best. Less clustered solutions imply that the reconstruction is more ambiguous.

To address the presence of several configurations, one should apply ensemble based tools such as MES or EOM. One could for example imagine a coexistence of strongly bent and extended confirmations of IgE Fc which could give the same SAXS signal as a moderately bent confirmation. Further, there is absolutely no point in discussing differences in chi-squared values. This can be easily understood that the same sample at lower concentration generally gives a lower chi-squared for the same model. A somewhat more meaningful estimator of fit-quality would be the p-value from the CORMAP test (Franke et al., 2014).

Page 11, 2nd paragraph, discussion of SAXS data: Following the above, it would be more appropriate to state that the data can be described by such models, as you cannot exclude the existence of other solutions.

Page 13, Acknowledgements:

I would strongly recommend to follow the acknowledgment guidelines for the beamlines used for this study:

<http://www.diamond.ac.uk/Beamlines/Mx/Common/Common-Manual/Acknowledging-MX-beamlines.html>

<http://www.esrf.eu/UsersAndScience/UserGuide/Preparing/GuidelinesToUsers>

Page 16, 2nd paragraph (Affinity measurements): The model underlying the KD calculation should be specified.

Page 18, 2nd paragraph (Small angle X-ray scattering data collection and rigid body modeling):

In the first line, 026 probably refers to sdab 026?

The structure of the data analysis section is very confusing. If the authors really used CRY SOL3 to calculate the scattering curves of the models generated by CORAL, this step is not only unnecessary but possibly also harmful, as CRY SOL does not necessarily treat regions added by CORAL correctly. Chi-squared values and predicted curves should be taken directly from CORAL. It is also not clear whether parts of the proteins missing in the crystal structures (Linker, termini) were explicitly modeled. This is the strong point of CORAL, otherwise SASREF could have been used.

Generally, for the publication of SAXS data, it is recommended to include a table listing "Data - collection and scattering-derived parameters" (<http://scripts.iucr.org/cgi-bin/paper?S0907444912012073>). The authors should include such a table. In addition, it is always a good idea to show Krakty plots (in the Supplement).

Missing literature:

Page 3, 3rd paragraph, the statement about the different bent and extended structures requires some backing.

Page 13, 3rd paragraph (Crystallization, data collection, structure determination and refinement):

Evans, G., et al. "Diamond Beamline 124: A Flexible Instrument for Macromolecular Micro-crystallography." AIP Conference Proceedings. Eds. Jae-Young Choi, and Seungyu Rah. Vol. 879. No.

1. AIP, 2007.

Page 18, 2nd paragraph (Small angle X-ray scattering data collection and rigid body modeling):

Pernot, Petra, et al. "Upgraded ESRF BM29 beamline for SAXS on macromolecules in solution." *Journal of synchrotron radiation* 20.4 (2013): 660-664.

Manuscript NCOMMS-17-13455A

Jabs et al., Trapping IgE in a closed conformation by mimicking CD23 binding prevents and disrupts interaction with FcεRI

Point by point reply to the reviewers' comments:

Reviewer #1 (Remarks to the Author):

There does not seem to be any data regarding the kinetics of disassociation of IgE from FcεRI due to the effects of the single domain antibody.

Response: We have addressed this point by including time course experiments (please refer to next point)

Another issue is that the data shown regarding human basophil tests in figure 4 seem to be derived from only one patient and none of these functional assays show relevant controls or statistics. Given differences in IgE sensitization and the functional sensitivity of basophils from donor-to-donor it would be useful to quantify the efficacy and potency of the single domain antibody at inhibiting IgE-dependent functional responses and IgE sensitization over a range of basophil donors.

Response: The reviewer is correct, the basophil analysis has been performed using exemplary one patient sensitized to birch pollen.

We consider the functional data shown in Figure 4 mainly as initial data that support the structural data on the IgE Fc conformation and suggest that the sdab exhibit effects on free IgE and IgE on effector cells.

Much broader studies on efficacy of the sdab will be needed to assess its potential as anti-IgE biological. Such broad analyses however need to be addressed in future studies.

We agree however with the reviewer and appreciated the suggestion that analysing a range of basophil donors is highly interesting and could provide additional, more quantitative insights into the effects.

To address this point we carefully recruited a panel of allergic patients without prior specific SIT. In addition to serological data we determined changes of surface IgE and basophil activation upon incubation with the sdab. For control purposes we also expressed, purified and included an inactive sdab mutant.

Now, analyses of surface IgE were performed using basophils from patients sensitized to birch pollen, yellow jacket venom, and honeybee venom and the experiments were done in a concentration and time-dependent manner.

For analysis of basophil activation we focused on birch pollen allergic patients since tree pollen allergy is an excellent model and the vast majority of patients react to the dominant component Bet v 1. In other allergies the sensitization profiles can be much more complex rendering a detailed analysis more difficult.

Quantification of the data obtained in BAT was performed by CDsens analysis, a well-established measure of basophil sensitivity threshold.

We are very pleased that the new data we obtained support and even extend our original data.

Overall the new data show even more impressively the reduction of surface IgE for different allergic basophil donors and thereby the displacement of IgE from FcεRI by the sdab. The data also underline that

displacement is a fast process at the sdab concentrations used here taking place within minutes. Prolonging the time to up to 60 min maximizes the effect, but only slightly.

Similarly the reduction of allergen-dependent basophil activation by the sdab could be observed for a panel of birch pollen sensitized donors having different levels of sIgE and tIgE. The reduction of CDsens for all patients analysed clearly documents that the displacement of surface IgE translates into reduced responsiveness of effector cells.

The new data has been introduced into the manuscript into the figures as novel panels in figure 4, supplementary figure S7 and S8 and as supplementary data table 1. The obtained results are described and further discussed in the results section and the discussion section.

The results section now reads as follows:

“Flow cytometry analysis of surface IgE on human basophils obtained from three allergic patients sensitised to inhalative and injected allergen showed that treatment with the O26 sdab for 15 min reduced the amount of surface IgE to approx. 30% (Fig. 4B, Table S1). Prolonging the incubation time further reduced surface IgE to approx. 20%. Neither the inactive sdab mutant 112 nor omalizumab exhibited comparable reduction. Incubation of immobilized IgE:FcεRI complexes with O26 sdab and omalizumab supported the displacement of IgE by the O26 sdab (Fig. S7).

Next we analysed the impact on basophil activation for a panel of 6 patients with birch pollen allergy (Table S1). The high prevalence of birch pollen allergy and the fact that birch pollen major allergen Bet v 1 is the predominant allergen for the vast majority of birch pollen allergic patients in Northern Europe allows for a good comparability of obtained results. Therefore we analysed changes in basophil allergen threshold sensitivity, CDsens, to Bet v 1 in basophil activation tests. CDsens is a well-established measure to quantify effector cell sensitivity to an allergen and reflects the amount of allergen needed for efficient activation. In accordance with the reduced surface IgE, the O26 sdab significantly reduced CDsens for patient’s basophils by 50-95%. (Fig. 4C-E). Levels of specific IgE (sIgE) to Bet v 1 and total IgE (tIgE) in serum suggest that lower concentrations of sIgE ans as a consequence thereof smaller ratios of sIgE to tIgE translate into a more efficient reduction of CDsens (Fig. S8). These findings are in line with the observation that polysensitisation is often accompanied with a less severe phenotype of the allergic response.”

The discussion section now reads as follows:

“Apparently, the sdab removes IgE from the surface of effector cells as shown for basophils obtained from patients with different types of allergies. The reduction of IgE resulting correlates within a decreased sensitivity of the effector cells to allergen-dependent activation as shown in a cohort of birch pollen sensitized patients. It is imperative to consider that sensitivity of effector cells is shaped by the characteristics of surface IgE, e.g. ratio of specific and total IgE, affinity and repertoire complexity³⁹. Although we used the dominant birch pollen allergen, Bet v 1, patient-specific sensitization profiles influence the reduction of cellular sensitivity, an effect likely to be even more pronounced in multisensitized individuals.”

Minor point: Page 11, line 8: “CD23 contacts residues in....” Change to “contact residues”?

Response: We reworded the sentence to “CD23 is contacting residues in ...”

Reviewer #2 (Remarks to the Author):

Specific comments:

**p. 4, in the introduction please describe sdab 026- was it from a camel or llama, or is it a human VHH?
How was it isolated?**

Response: We agree that more information about the sdab is needed and introduced additional information on the sdab into the introduction section.

The section now reads as follows:

“Here we report the crystal structure and structure in solution of the human IgE Fc in complex with the sdab 026, a llama-derived, humanized sdab selected against IgE.”

p. 6, please give a better description of the ‘hepta-saccharide’- was this a Nag2Man4?

Response: We added information about the heptasaccharide to the text:

“In our structure we observe this as a (N-acetylglucosamine)₂(mannose)₅ (NAG₂MAN₅) hepta-saccharide, which ...”

p. 13, in the materials and methods, hopefully your crystals were ‘cryocooled’, not frozen.

Response: The wording has been corrected.

In Table 1, please include Rmerge and Rpim. Under the refinement statistics, what are the ligand atoms? I would specify as water or carbohydrate or whatever they are to be more clear.

Response: The missing information has been added and the wording has been specified.

In the Figure 3 legend, please include the PDB code for the Fc-FcεRI complex shown in panel D.

Response: The PDB code has been added as requested.

Reviewer #3 (Remarks to the Author):

Parts of the introduction read rather awkwardly, it might profit from thorough proofreading by an outsider. For example in “The human FcεRI is expressed as a tetramer or a trimer lacking the signal amplifying β-subunit.” It is not clear whether only the trimer or both the trimer and the tetramer lack the subunit.

Response: In order to improve readability we reworded sentences, phrases and wording in the introduction. We hope to have improved readability here.

Page 5, 1st paragraph, analysis of the surface plasmon resonance experiment: The combination of a KD value with a χ^2 is nonsensical. The KD should have a margin of confidence, whereas χ^2 is a property of the fit (and often meaningless).

We agree with the reviewer and removed χ^2 . A short statement and a reference to available information have been added.

Page 9, SAXS analysis: It is not possible to draw any conclusion of the ensemble of solution structure from CORAL modeling. Clustering of the results only imply that the reconstruction is stable, but CORAL by design will always find the SINGLE model that describes the data best. Less clustered solutions imply that the reconstruction is more ambiguous.

To address the presence of several configurations, one should apply ensemble based tools such as MES or EOM. One could for example imagine a coexistence of strongly bent and extended conformations of IgE Fc which could give the same SAXS signal as a moderately bent conformation.

Further, there is absolutely no point in discussing differences in chi-squared values. This can be easily understood that the same sample at lower concentration generally gives a lower chi-squared for the same model. A somewhat more meaningful estimator of fit-quality would be the p-value from the CORMAP test (Franke et al., 2014).

Response: We agree on the comments regarding the interpretation of clustering within rigid body models, and we have removed the prior statements hinting at a biological meaning of tight/loose clustering in the CORAL output models. Also we have removed comparisons of the chi2 values after CORAL modelling, we now merely provide the values. The reviewer’s suggestion of attempting ensemble optimization is highly relevant considering the known flexibility of IgE Fc and we now describe EOM analysis for both unbound IgE Fc and the Fc-sdab complex. In both cases two representative models were output with the major conformation closely resembling the models from CORAL rigid body modelling while a second minor conformation is somewhat different. Overall the conclusions from CORAL and EOM are the same, unbound IgE Fc appears to adopt mainly the bend conformation in solution whereas in the sdab complex a novel less bend conformation appears to be the dominating conformation. The entire SAXS paragraph has been extensively rewritten, the presentation of the rigid body modelling has been truncated, the EOM analysis has been added, and a new SAXS data table requested by the reviewer has been added.

Page 11, 2nd paragraph, discussion of SAXS data: Following the above, it would be more appropriate to state that the data can be described by such models, as you cannot exclude the existence of other solutions.

Response: We agree and now write in the discussion “Being unable to crystallize sdab in complex with the full IgE Fc we used SAXS to analyze the complex and free IgE Fc in solution. This allowed for the first time insights into the positioning of the Cε2 domain in solution for both the free Fc as well as in the complex with the sdab. Two different approaches to SAXS based modelling suggested that with respect to the sdab complex, the IgE Fc Cε2 domains appear to adopt a major conformation in between the two well established bent and extended IgE Fc conformations. In unbound IgE Fc, the dominating conformation appears to be the bend conformation. In addition, EOM analysis suggested that one minor conformation somewhat different from the major conformation may be present in both cases.”

Page 13, Acknowledgements:

I would strongly recommend to follow the acknowledgment guidelines for the beamlines used for this study:

<http://www.diamond.ac.uk/Beamlines/Mx/Common/Common-Manual/Acknowledging-MX-beamlines.html>

<http://www.esrf.eu/UsersAndScience/UserGuide/Preparing/GuidelinesToUsers>

Response: We appreciate the suggestion and modified the acknowledgements accordingly.

Page 16, 2nd paragraph (Affinity measurements): The model underlying the KD calculation should be specified.

The information was added as suggested.

Page 18, 2nd paragraph (Small angle X-ray scattering data collection and rigid body modeling): In the first line, 026 probably refers to sdab 026?

The structure of the data analysis section is very confusing. If the authors really used CRY SOL3 to calculate the scattering curves of the models generated by CORAL, this step is not only unnecessary but possibly also harmful, as CRY SOL does not necessarily treat regions added by CORAL correctly. Chi-squared values and predicted curves should be taken directly from CORAL. It is also not clear whether parts of the proteins missing in the crystal structures (Linker, termini) were explicitly modeled. This is the strong point of CORAL, otherwise SASREF could have been used.

Response: We now give the chi2 values output from directly from CORAL rather than with CRY SOL3 and we now also give the comparison between experimental and calculated data as output from CORAL in Fig 5B and 5D. To comfort the reviewer there are no notable differences to the chi2 values and predicted scattering curves from CRY SOL3, but we agree that using the CORAL output is a more stringent approach.

Generally, for the publication of SAXS data, it is recommended to include a table listing "Data-collection and scattering-derived parameters" (<http://scripts.iucr.org/cgi-bin/paper?S0907444912012073>). The authors should include such a table. In addition, it is always a good idea to show Kratky plots (in the Supplement).
Response: We did already show the Kratky plot in the original manuscripts as Fig S10D. The requested table is now included as table S2

Missing literature:

Page 3, 3rd paragraph, the statement about the different bent and extended structures requires some backing.

Page 13, 3rd paragraph (Crystallization, data collection, structure determination and refinement):
Evans, G., et al. "Diamond Beamline 124: A Flexible Instrument for Macromolecular Micro-crystallography." AIP Conference Proceedings. Eds. Jae-Young Choi, and Seungyu Rah. Vol. 879. No. 1. AIP, 2007.

Page 18, 2nd paragraph (Small angle X-ray scattering data collection and rigid body modeling):
Pernot, Petra, et al. "Upgraded ESRF BM29 beamline for SAXS on macromolecules in solution." Journal of synchrotron radiation 20.4 (2013): 660-664.

Response: We agree that the mentioned literature should be added and updated the references accordingly.

REVIEWERS' COMMENTS:

Reviewer #3 (Remarks to the Author):

I would like to thank the authors for the efforts they invested into improving the manuscript. It was a very pleasant read.

I only have a few very minor comments that should be easily addressed:

Page 20 ("Small angle scattering data collection and modeling")

"Calculation of theoretical scattering profiles of atomistic structures from CORAL modeling and their fits to the experimental data (as measured by the chi2 value) were done using CRY SOL3."

In your response you state

"We now give the chi2 values output from directly from CORAL rather than with CRY SOL3 and we now also give the comparison between experimental and calculated data as output from CORAL in Fig 5B

and 5D. To comfort the reviewer there are no notable differences to the chi2 values and predicted scattering

curves from CRY SOL3, but we agree that using the CORAL output is a more stringent approach."

But this is not how I would understand the above sentence - I would naively assume that all chi2 values come from CRY SOL3.

Perhaps a formulation along these lines would be less misinterpretable:

"Fits of atomistic models to the experimental data (as measured by the chi2 value) were done using CRY SOL3 and atomistic models to fit the data were build using CORAL."

Figure 5B,D; Supplement Figure S10A,B,C:

The y-axis deserves some units. If your data comes from BM29, I suspect these might be kDa mg/mL. If your data was scaled, the units are of course arbitrary, but that could explicitly stated.

Supplement Table S2:

If BSA is used as a standard, the exact product and supplier should be specified, as well as how the sample was prepared (buffer, concentration,...).

72 kDa are not the molecular weight of BSA, does this value take dimers and/or contaminants into account? Additionally, for data collected at ESRF BM29 you should have a water measurement. At higher angles, water scatters at 0.0163/cm, which can be used to calculate the scaling.

Supplement Figure S10B,C:

it would be nice if you could show how the data continues outside of the Guinier range.

Point by point reply Jabs et al., MS-17-13455A

REVIEWERS' COMMENTS:

Reviewer #3 (Remarks to the Author):

Page 20 ("Small angle scattering data collection and modeling"): "Calculation of theoretical scattering profiles of atomistic structures from CORAL modeling and their fits to the experimental data (as measured by the chi2 value) were done using CRY SOL3."

In your response you state:

"We now give the chi2 values output from directly from CORAL rather than with CRY SOL3 and we now also give the comparison between experimental and calculated data as output from CORAL in Fig 5B and 5D. To comfort the reviewer there are no notable differences to the chi2 values and predicted scattering curves from CRY SOL3, but we agree that using the CORAL output is a more stringent approach." But this is not how I would understand the above sentence - I would naively assume that all chi2 values come from CRY SOL3.

Perhaps a formulation along these lines would be less misinterpretable:

"Fits of atomistic models to the experimental data (as measured by the chi2 value) were done using CRY SOL3 and atomistic models to fit the data were build using CORAL."

Answer: All chi2 values given in the revised MS were from CORAL, CRY SOL3 was not used any longer. Unfortunately the statement concerning CRY SOL3 mentioned by the reviewer was not deleted from the revised MS. It has now been removed.

Figure 5B,D; Supplement Figure S10A,B,C:

The y-axis deserves some units. If your data comes from BM29, I suspect these might be kDa mg/mL. If your data was scaled, the units are of course arbitrary, but that could explicitly stated.

Answer: We agree and modified these panels accordingly with the unit kDa, which has been confirmed by BM29 staff. In addition we have added the unit (nm⁻²)/kDa to the Kratky plot in Fig S10D.

Supplement Table S2:

If BSA is used as a standard, the exact product and supplier should be specified, as well as how the sample was prepared (buffer, concentration,...).

Answer: We have added to the methods section "Bovine serum albumin (Sigma, A7638-5GP) used for calculation of the molecular weight (Suppl table 1) was solubilized in a buffer containing 50 mM Hepes pH 7.5 at concentration of 5 mg ml⁻¹."

72 kDa are not the molecular weight of BSA, does this value take dimers and/or contaminants into account? Additionally, for data collected at ESRF BM29 you should have a water measurement. At higher angles, water scatters at 0.0163/cm, which can be used to calculate the scaling.

Answer: The used molecular weight for BSA accounts for the monomer/dimer equilibrium of BSA as described in ¹.

There is indeed a water measurement available, which was used by the data processing pipeline for scaling. We chose to estimate the molecular weight in Suppl table 2 by comparison to the BSA standard,

which is a known to be a reliable procedure and comparable to calibration against a water measurement².

- 1 Petoukhov, M. V. *et al.* Reconstruction of quaternary structure from X-ray scattering by equilibrium mixtures of biological macromolecules. *Biochemistry* 52, 6844-6855, doi:10.1021/bi400731u (2013).
- 2 Mylonas, E. & Svergun, D. I. Accuracy of molecular mass determination of proteins in solution by small-angle X-ray scattering. *J Appl Crystallogr* 40, S245-S249, doi:10.1107/S002188980700252x (2007).

Supplement Figure S10B,C:

it would be nice if you could show how the data continues outside of the Guinier range.

Answer: The Figure S10B-C has been modified in order to show additional points at both lower and higher values of q not used for fitting the straight line.